# A Key to Stimulate Green Technology Innovation in China: The Expansion of High-Speed Railways

**DOI:** 10.3390/ijerph20010347

**Published:** 2022-12-26

**Authors:** Ziyang Chen, Xiao Feng, Ziwen He

**Affiliations:** School of Economics and Management, Tongji University, Shanghai 200082, China

**Keywords:** high-speed railways, green technology innovation, creative class flow, environmental concerns, spatial spillover effect

## Abstract

Mankind is seeking a green development path. This paper places emphasis on whether high-speed railways (HSRs), as a typical representative of green transportation, can effectively promote green technology innovation in cities. Based on the panel data of 286 Chinese prefecture-level cities from 2007 to 2018, we employ the Panel Negative Binomial Regression Model and the Spatial Dubin Model for empirical analysis. The results illustrate that the expansion of HSRs not only has a direct and substantial promotion influence on local green technology innovation but also on the surrounding area. We further find that circulation node cities reap more benefits of the opening of HSRs than other ordinary cities. The higher the degree of marketization, the weaker the marginal impact of HSRs on green technology innovation. Meanwhile, the mechanism test confirms that HSRs can indirectly stimulate the progress of green technology innovation by influencing the creative class flow and the government’s environmental concerns. Our findings present new insights for enhancing green technology innovation and provide policy recommendations for local governments to take advantage of HSRs to obtain resources.

## 1. Introduction

Since the birth of the 21st century, countries around the world have increasingly paid attention to ecological and environmental issues. The demand for green economic growth has become increasingly urgent. The Chinese government is no exception, aiming to accomplish carbon peaking by 2030 and carbon neutrality by 2060 [1]. Therefore, they are actively seeking a green development path. Green technological innovation can simultaneously advance energy efficiency and diminish environmental pollution. It has become an effective approach for countries to accomplish green and sustainable development [2,3,4,5]. Hence, how to effectively accelerate green technology innovation has turned out to be a topic worth studying [6,7].

High-speed railways (HSRs), as a new representative of rapid transportation, not only promote the efficient progress of the economy and society [8,9] but also have a positive and profound impact on the advance of both knowledge spillover and traditional innovation [10,11,12]. Nevertheless, only a small number of scholars have deeply explored the causal association and influence mechanism between HSR development and green technology innovation. Existing studies either focus on factor flows or focus on local governments’ behaviors [13]. In addition, some studies only have theoretical analysis without rigorous empirical support [14]. As a result, this paper strives to narrow the gap in this kind of research. Meanwhile, as China has the largest HSR network, this paper utilizes the panel data of Chinese prefecture-level cities between 2007 and 2018 to conduct an empirical test on the impact of HSRs on green technology innovation.

The main contributions of this paper are as follows: Firstly, in order to further explore how HSRs lead green technology innovation, this paper not only analyzes from the perspective of creative class mobility but also proposes that government environmental concern is another important mechanism. Secondly, this paper uses the government work report to construct an indicator of environmental concern. Few studies have employed such data. Past studies have only been able to measure environmental regulation through pollutant indicators. Thirdly, this paper conducts a heterogeneity analysis on samples from location conditions and marketization degree. The influence of HSRs on various kinds of cities can be estimated more accurately. Finally, this paper applies the Spatial Dubin Model (SDM) to further measure the impact of HSRs on green technology innovation in surrounding cities from the standpoint of spatial spillover. It helps to overcome the problem that other studies ignore spatial factors.

The remaining sections are arranged as follows: The second part is the literature review, followed by the mechanism analysis and research hypothesis. The fourth part is method and data. The fifth part is empirical analysis, while the final part is the conclusion and future direction.

## 2. Literature Review

Presently, many academics have established the effectiveness of government policies in promoting the capacity of green technology innovation [15,16,17]. Some studies confirm that incentive-and-support policies can alleviate market failure in R&D activities and stimulate green technology innovation [18,19,20,21,22,23], for example, providing R&D subsidies or tax breaks, opening green credit channels, and establishing emission trading markets. Other studies point out that command-and-control policies are more stimulating [24,25,26]. For example, levying pollutant discharge fees and environmental taxes, issuing pollutant discharge permits, setting pollutant discharge standards, and publicizing enterprises’ environmental reports touch on the influence of green technology innovation. Additionally, the background, timing, and intensity of policy implementation also shake the effect of green technology innovation [27,28,29]. However, it should be emphasized that most of the literature on the impact of policies on green technology innovation is focused on fiscal, financial, and environmental aspects. Few scholars have explored its incentive force on green technology innovation from the perspective of transportation expansion. The development of transportation has always been a critical approach to promoting economic and social advancement because the knowledge spillover and diffusion brought by talent flow is an essential mechanism to improve the innovation level [30,31]). Mobility depends largely on transportation infrastructure. The construction of transportation infrastructure can diminish the expenditure on travel and enhance the possibilities of face-to-face communication [32]. Therefore, it is meaningful to explore the connection between HSR development and green innovation.

Most articles about the HSRs and innovation focus on traditional technological innovation. The literature from the regional level generally believes that the expansion of the HSRs can significantly advance the regional innovation level [10,11]. With regard to both quantity and quality, the regions where HSRs have opened have shown significant growth in innovation productivity [33]. Some scholars have found that HSRs can greatly improve cross-city collaborative innovation capacity [34,35]. However, this incremental research collaboration essentially takes place between the core city and the surrounding city in the innovation system [36]. Similar conclusions have been drawn from the enterprise-level literature. That is, the initiation of HSRs can not only considerably improve the innovation capacity of enterprises along the line [37,38,39], but also promote collaborative innovation among enterprises [40]. Moreover, many scholars have emphasized that technological innovation is the fundamental way for HSRs to improve energy efficiency and reduce pollution emissions [41,42,43,44,45]. As stated by the World Intellectual Property Organization [46], the technologies associated with pollutant disposal and climate change mitigation are precisely green technologies. It can be seen that there is a direct and close association between HSR construction and green technology innovation.

Different from traditional innovation, green technology innovation can not only improve productivity but also reduce pollution and pollution control costs, which is the key to balancing economic progress and environmental conservation [4]. Consistent with our knowledge, there are a few articles that directly study the connection between HSRs and green technology innovation. Huang and Wang [13] suggested that HSRs enhance the possibility of regional green innovation activities by enhancing the mobility of high-tech talents. Zhu et al. [47] proposed that HSRs expedite the movement of innovation factors, including labor and capital. Moreover, they claimed that the calculation of factor flows should consider both inflows and outflows. Yu and Wan [14] theoretically demonstrated that HSRs can also improve local green innovation by influencing government behavior. However, they lacked strong and rigorous empirical analysis. Lin et al. [48] found that HSR service speeds the growth of green technology innovation by furthering the cooperative agglomeration of manufacturing and producer services. Therefore, this paper aims to enrich this part of the research.

In addition, many studies have revealed that HSRs have a significant spatial spillover effect [42,43,49,50], influencing the model of regional economic development [51]. Recently, Zhou et al. [52] theoretically summarized three driving paths for regional green technology innovation spillover under the power of HSRs by using the heat conduction model in physics. They found that HSRs have the advancement effect of opening HSRs on the spillover of regional green technology innovation. However, they offer no empirical support. Based on this, it is necessary to further investigate the impact of HSRs on green technology innovation in local and surrounding cities by adding spatial factors.

## 3. Mechanism Analysis and Research Hypothesis

As a fast and environmentally friendly means of transportation, HSRs are not only convenient for people to travel but also significantly affect economic development and the natural environment [42,53]. Coupled with the investigation of the existing literature, this paper believes that HSRs exert an impact on green technology innovation mainly through two important mechanisms: creative class mobility and the government’s environmental concern. The conceptual diagram of this article is displayed in Figure 1.

The stream of innovation factors, including the flow of the creative class, is the core of technological innovation [54]. For the creative class, transportation convenience is one of the critical elements shaping their agglomeration [55]. Meanwhile, many studies have pointed out that face-to-face interaction is the secret to realizing localized knowledge spillover and hidden knowledge exchange [56,57]. HSRs reduce people’s commuting costs and time while accelerating the flow of innovative talent across regions [33,34,58]. It not only helps to accelerate the dissemination of knowledge among different groups but also stimulates the formation of new knowledge. Specifically, the creative class can form a moderate concentration in some cities in a short period. It enables the timely sharing of knowledge related to green technologies. Then, the newly generated knowledge and green technologies can spread quickly and efficiently with the HSR network [59]. Furthermore, the improvement in transportation convenience initiated by HSRs will increase the possibility of face-to-face communication among creative talents [11,36,59,60,61]. The decrease in communication costs will indirectly induce scientific research cooperation [33], leading to the improvement of green technology innovation. In addition, the flow of creative talents brought about by HSRs can also be seen as the flow of consumers. When consumers have more product choices, companies in the market pool will face greater competition. In order to increase market share and reduce pollution control costs, enterprises will improve their market competitiveness through green innovation [62]. Hence, the flow of the creative class between different cities can trigger the production and spillover of green technology and drive the innovation of the whole region.

Many studies have shown that HSRs can increase passenger volume of station cities and promote the expansion of tourism in cities along the railway [63,64]. As a result, the city connected by HSRs will come into the public eye, which will motivate the government to improve the environmental conditions and create a good image of the city. The government’s concern for the environment can sway green technology innovation [65,66,67]. The government is the core organizer and manager of social affairs. The government has the initiative to enforce environmental regulations only when it pays high attention to the environment [68]. Properly used, environmental governance can force a green transformation of polluting industries [69,70,71,72,73]. Meanwhile, in the context of innovation competition among local governments in China, the government is a stakeholder in green technology innovation [74]. The government, which values the environment and innovation, is more willing to increase investment in special funds for environmental protection [75]. It can not only ease the fiscal constraints of innovative enterprises but also provide a financial guarantee for green technology R&D activities [76]. Different from traditional innovation, consumers have a low readiness to subsidize the premium of green products. Consumer demand is not enough to encourage green technology innovation [77,78]. As a result, government policies are needed to awaken public consciousness of environmental conservation and increase market demand for green technologies and products [70,79,80,81]. Thus, enterprises and research institutions will be stimulated to pursue green technology and boost spending in green technology innovation. Therefore, HSR construction effectively promotes green technology innovation by raising the government’s environmental concerns.

Additionally, HSRs have an obvious network externality [82]. Consequently, the impact of HSRs on green technology innovation is no longer limited to local areas but will spill over to surrounding areas [49]. On the one hand, the impact of HSRs on green technology innovation may have a positive spillover effect [83]. That is, the boost power of HSRs on local green technology innovation will spill over to neighboring cities and drive green innovation in neighboring cities. On the other hand, HSRs may also generate the siphon effect [84]. In other words, the green innovation resources of disadvantaged cities along the HSRs are attracted to the central cities of the region, thus inhibiting green technology innovation in these cities.

Based on preceding discussion, this paper suggests the following hypotheses:

**H1.** 
*The opening of HSRs can effectively foster green technology innovation in cities.*


**H2.** 
*The opening of HSRs can stimulate green technology innovation in cities by boosting the creative class flow.*


**H3.** 
*The opening of HSRs can increase the environmental concerns of local government, thus effectively promoting green technology innovation.*


**H4.** 
*HSR development has a significant spatial spillover effect on green technology innovation.*


## 4. Methods and Data

### 4.1. Econometric Model

#### 4.1.1. Baseline Regression Model

The annual quantity of green patent applications in cities is considered to be a surrogate variant of green technology innovation, which belongs to counting data. Therefore, this paper uses panel Poisson regression to estimate the connection between HSRs and green technology innovation. However, considering the excessive dispersion of the amount of green patent applications, the panel negative binomial regression model was selected. The explicit regression model is as follows:(1)$$E\left[{\mathrm{GTI}}_{i,t}\right]=\exp\left(\alpha_{0}+\alpha_{1}{\mathrm{HSR}}_{i,t}+\alpha_{2}X_{i,t}+\mu_{i}+\gamma_{t}+\varepsilon_{i,t}\right)\;$$
where i is the city, t is the year, $E\left[{\mathrm{GTI}}_{i,t}\right]$ is the expected value of green technology innovation in year t of the city I, HSR is the dummy variable of HSR, X is a set of control variables, $\mu_{i}$ and $\gamma_{t}$ are city fixed effect and year fixed effect, respectively, and $\varepsilon_{i,t}$ is the random error term.

#### 4.1.2. Spatial Spillover Effect Model

With regard to the articles of Yu et al. [49] and Zhang et al. [43], the SDM was introduced to verify the spatial spillover effect of HSRs on green technology innovation. The specific model is established as follows:(2)$${\mathrm{GTI}}_{i,t}=\rho \sum_{j=1}^{N}w_{\mathrm{ij}}{\mathrm{GTI}}_{i,t}+\beta_{1}{\mathrm{HSR}}_{i,t}+\eta_{1}\sum_{j=1}^{N}w_{\mathrm{ij}}{\mathrm{HSR}}_{i,t}+\beta_{2}X_{i,t}+\eta_{2}\sum_{j=1}^{N}w_{\mathrm{ij}}X_{i,t}+\mu_{i}+\varepsilon_{i,t}$$
where ${\mathrm{GTI}}_{i,t}$ applies the logarithm of green patent applications per capita as a proxy variable, $w_{\mathrm{ij}}$ is an element of the spatial weight matrix $W_{1}$, and $\mu_{i}$ is city fixed effect. The other variables are consistent with the above.

This paper builds a spatial weight matrix ($W_{1}$) based on geographic distance, and the explicit element of $W_{1}$ is set as follows:(3)$$w_{\mathrm{ij}}=\left\{\begin{array}{c} \frac{1}{{d_{\mathrm{ij}}}^{2}},i\neq j\\ 0,\;i=j \end{array}\right.$$
where $d_{\mathrm{ij}}$ represents the geographical distance between two cities, intended based on the longitude and latitude of the city.

### 4.2. Selection and Calculation of Variables

#### 4.2.1. Dependent Variable

At present, scholars mainly use two methods to evaluate green technology innovation. The first method is to directly assess the number of green patents [85,86], although the number of patents is an absolute value. Green patents are selected from all patents as stated by the International Patent Classification (IPC) Green List published by the IPC Expert Committee. They are associated with environmentally sound technologies. The second method is to add environment-related factors into total factor productivity. This part of the literature usually employs the method of undesired output super-SBM (slacks-based measure of super efficiency) model combined with the GML (Global Malmquist–Luenberger) index [87,88]. After considering the input of labor capital and energy in research and development, they take GDP or the number of patents as expected output variables and pollutant emission as unexpected output variables. However, this paper argues that this method does not pay heed to the cost of environmental governance and the economic loss caused by environmental pollution. It cannot truly reflect the relationship between the input and output of green technological innovation. On the contrary, green patents are direct products of green technological innovation and can effectively reflect the degree of research and innovation in the year. Meanwhile, considering that there is a long period of time between patent application and authorization, the amount of green patent applications is finally employed as the surrogate variable of green technology innovation (GTI) in this paper.

#### 4.2.2. Independent Variable

HSRs are a valuable type of transportation infrastructure in China today, with a designed speed of 250 km per hour or more. As of 2021, China had more than 40,000 km of HSRs in operation, ranking first in the world [89]. By June 2022, the total mileage of the normal operation at 350 km per hour reached nearly 3200 km [90]. Based on the accessibility of data, most studies assume the introduction of HSRs as a quasi-natural experiment and set whether the city has opened the HSR service as a dummy variable. Then, they utilize the differential-in-difference model to test the economic consequences of HSRs [44,91]. Some researchers tend to use the index of accessibility to highlight the advantages of space–time compression brought by HSRs [43,92]. Some articles obtain the number of HSR stops by collecting and sorting the HSR information table manually [42,48,93]. Considering the availability and timeliness of data, this paper uses a dummy variable to represent HSRs. If HSRs are equal to 1, HSRs have been opened. If the value is 0, no HSRs are in operation.

#### 4.2.3. Control Variable

For reducing the endogeneity of the model, this paper adds some additional indicators as control variables, including economic development (ED), industrial structure (IS), government support (GS), financial development (FD), foreign investment (FI) and other transportation (otherVol).

GDP per capita is employed to reflect the degree of economic progress (ED). Rich cities mean relatively lower pressure on economic development, and they have greater motivation for environmental protection and financial security and can devote themselves to green development [94]. Therefore, economic development would significantly promote green technology innovation.

Tertiary industries, especially high-tech industries, are not only energy-efficient and clean industries but also important sources of technological innovation [95]. Scientific allocation of the proportion of tertiary industry can encourage the innovation of green technology and achieve the effect of green economic transformation. This paper adopts the proportion of the added value of tertiary industry and secondary industry as the proxy variable of industrial structure (IS).

The government’s support for scientific and technological innovation has a direct incentive effect on green technological innovation [96]. Financial grants from the government can effectively relieve the financing pressure on innovative enterprises and research institutes and guarantee the smooth progress of R&D activities. This paper employs the ratio of fiscal expenditure on science to GDP to measure government support (GS) for innovation activities of green technology.

The essence of finance should be similar to lubricant, which can help enterprises solve financing problems, stabilize their financial status, and facilitate enterprises’ green technology innovation activities [21]. Financial development (FD) is assessed by the percentage of total deposits and loans of financial institutions to GDP at the end of the year.

Reviewing the literature, there are both positive and negative views on the impression of foreign investment in green technology advances in host countries. Some scholars believe that foreign investment is beneficial to the development of green technology because of its significant technology spillover effect [97]. However, other scholars point out that foreign investment has no obvious effect on technological progress [98]. Under the background of the pollution paradise hypothesis, foreign investment does not drive the innovation of green technology in the host country. The proportion of FDI used in GDP is selected as a proxy variable of foreign investment (FI).

Moreover, HSR passengers can also travel by self-drive, bus, or plane. Therefore, the total passenger volume of road and air is added to control the influence of other modes of transportation (otherVol).

#### 4.2.4. Mechanism Variable

The creative class is the main body of innovation. As mentioned above, the travel convenience and cost reduction brought by HSRs will lead to the flow of the creative class. Due to the lack of actual data on the turnover of the creative class, the gravity model is used to value the potential flow of the creative class between cities [99]. The salary level of the outside cities will have different degrees of attraction for the local creative class, leading to the possibility of brain drain. The specific calculation formula of creative class flow (CCF) is as follows:(4)$${\mathrm{CCF}}_{\mathrm{ij}}=\frac{{\mathrm{CC}}_{i}\times {\mathrm{avWage}}_{j}}{R_{\mathrm{ij}}^{2}}$$
(5)$${\mathrm{CCF}}_{i}=\sum_{1}^{n}{\mathrm{CCF}}_{\mathrm{ij}}$$
where ${\mathrm{CCF}}_{\mathrm{ij}}$ represents the potential flow of the creative class from city i to city j; ${\mathrm{CC}}_{i}$ is the employment number of the creative class in city i; ${\mathrm{avWage}}_{j}$ is the average wage of an employee in city j, measuring the appeal of city j; $R_{\mathrm{ij}}$ is the rectilinear distance between city i and city j, computed based on the longitude and latitude of the city; and ${\mathrm{CCF}}_{i}$ is the sum of potential flows of the creative class in city i to all other cities. According to the description of the creative class by Florida [55] and the industry classification published by the China Statistics Bureau, this paper classifies employees in 10 industries as the creative class: information, finance, science and technology, education, culture, real estate, leasing, water conservancy, health, and public administration. It is worth noting that ${\mathrm{CCF}}_{\mathrm{ij}}\neq {\mathrm{CCF}}_{\mathrm{ji}}$, that is, the mutual attraction between cities is not equal.

As described above, the government’s emphasis on the environment will affect the intensity of green technology innovation in the region. Referring to the method of Chen and Chen [100], this paper uses the frequency of environmental words in government working statements as a proxy variable to measure environmental concern (EC). Specifically, the annual government working statements of prefecture-level cities are processed by word segmentation. The frequency of environment-related words (such as environmental protection, environmental pollution, energy consumption, emission reduction, sewage discharge, ecological, green, low-carbon, air, chemical oxygen demand, sulfur dioxide, carbon dioxide, PM2.5, etc.) was also totaled. The more frequently environment-related words appear in government working statements, the higher importance the government attaches to environmental problems.

### 4.3. Data

Considering the time of HSR construction and the availability of other associated data, this article picks the 286 prefecture-level cities in China as the research objects. Given the remarkable impact of the pandemic of COVID-19, the research period is set as 2007 to 2018. Most of the data related to the features of prefecture-level cities come from the *China Urban Statistical Yearbook* and *China Regional Economic Statistical Yearbook* [101]; Data on the number of patent applications came from the State Intellectual Property Office, and then green patents were screened out according to the IPC Green List. The HSR data mainly come from the China Research Data Service Platform (CNRDS). The data used for environmental concern came from government working statements of prefecture-level cities. For some missing data, the manual completion method and linear interpolation method were employed to supplement. To solve the problem of heteroscedasticity and zero value, this paper adds 1 to all variables and then takes logarithms of them (except for GTI and HSRs). Table 1 shows the results of descriptive statistics for variables mainly used in this paper.

## 5. Analysis of Findings

### 5.1. Baseline Regression Analysis

Before the baseline regression, all variables had passed the variance inflation factor (VIF) test, which confirmed the absence of multicollinearity. The rationale for using the fixed-effect model is substantiated by the Hausman test. Moreover, as shown in Table 1, the standard deviation of GTI is almost three times the mean value, indicating that the data may be excessively dispersed. Hence, applying the panel negative binomial regression model is more efficient than the Poisson regression model.

Columns (1) and (2) of Table 2 show the outcomes of baseline regression by employing the panel negative binomial model (NB)., According to Column (2), the introduction of HSRs increase the number of green patent applications in cities by 7.2% under a confidence level of 5%. Therefore, Hypothesis 1 of this paper is confirmed. That is, the operation of HSRs can appreciably foster the green technology innovation in cities. This is consistent with the conclusions of previous scholars, such as Zhu et al. [47] and Yu and Wan [14].

From the coefficients of control variables in Column (2) of Table 2, we further obtain that the level of economic development (ED) is directly proportional to the output of green technology innovation. At the 1% confidence level, a 1% increase in GDP per capita leads to a 41.3% expansion in green patent applications. It seems to confirm that economic advance will improve the awareness of green energy conservation, which contributes to the promotion of green technology [94]. The coefficient of industrial structure (IS) is positive at the confidence level of 1%. It indicates that the optimization of industrial structure contributes to the innovation of green technology. The tertiary industry is knowledge-intensive and has low energy consumption. The industrial structure is tilted towards the tertiary industry, which means that there is more human capital to support green innovation activities. So, it enhances the ability of green technology innovation. Meanwhile, there is a substantial positive association between government support (GS) and green technology innovation, which is consistent with Roh et al. [96]. It discloses that the government’s financial subsidies, on the one hand, send a signal to enterprises that they attach importance to green technological innovation. On the other hand, it provides financial security for the innovation activities of enterprises. The coefficient of financial development (FD) is positive but weakly meaningful. The probable reason is that money moves between financial institutions without finding its way into the real economy. Financial institutions have not been effective in alleviating the problem of corporate finance, thus failing to promote green technology innovation. In addition, the endorsement effect of foreign investment (FI) on green technology innovation is also not significant. It confirms the “pollution paradise hypothesis”, in which the infusion of foreign capital does not bring green technology innovation to the host country.

Columns (3) to (6) report the results of robustness tests. Specifically, Columns (1) and (2) are the results before and after adding control variables. Column (3) uses the lag term of the HSRs after considering the time lag effect of HSRs. Column (4) and Column (5) use the panel Poisson model (Poisson) and panel fixed effect model (FE), respectively, to estimate the results from the perspective of the replacement measurement method. Column (6) shows the regression results after excluding municipalities and provincial capitals from the sample (based on the panel negative binomial fixed effect model). A larger value of the Log likelihood denotes a better fit of the model. So, the negative binomial model is proved to be more suitable than the Poisson model. The Wald Chi2 is an indicator of the significance of a model. The *p*-value of Chi2 is 0.000, demonstrating that the model is rational. As demonstrated in Columns (1) to (6), the coefficient of the variable HSRs always remains positive at least at the confidence level of 10%. The result has been tested to be robust.

### 5.2. Endogenous Problem

Although the dummy variables of city and year have been controlled in the baseline regression, there may still be endogenous problems in the OLS estimation of green technology innovation of HSRs. There are two reasons: Firstly, the spatial layout of the HSR network and the location of the HSR station may be related to local development and government resources. Secondly, there may be missing variables. Therefore, this paper employs two methods to address the issue of endogeneity.

Using instrumental variables is one of the effective ways to solve endogeneity problems. Therefore, this paper draws on the method of Yang et al. [102] to construct instrumental variables from the perspective of geography and climate. Specifically, the average urban slope is used as the instrumental variable of HSRs. Urban slope satisfies exogenous characteristics and reflects the construction difficulty and economic cost of HSR construction. So, it meets the requirements of instrumental variables. However, considering that the average slope is sectional data, this paper multiplies it with the dummy variable of the year to construct the panel instrumental variable. Column (1) of Table 3 discloses the estimation results using instrumental variables combined with the GMM model (IV-GMM). The outcomes of the unidentifiable test, weak instrumental variable test, and overidentification test (Hansen J statistic) confirm the validity of the instrumental variable.

The lag-terms of explained variables are introduced to form a dynamic panel, considering that the current status of green technology innovation may be disturbed by the past situation. Then, the system GMM estimation method (SysGMM) is used. Column (3) in Table 3 shows that the difference in disturbance term has first-order autocorrelation but no second-order autocorrelation. The outcomes of the Hansen test reveal that the overidentification test can be passed at the significance level of 10%. Meanwhile, the *p*-value of the Wald test is 0.000, indicating that the model is significant. Therefore, it is feasible to build dynamic panels and utilize the system GMM model to unravel the endogeneity puzzle.

To sum up, after addressing the endogenous issue, the chief denouement of this paper is robust and reliable. HSR construction can raise the standard of green technology innovation.

### 5.3. Heterogeneity Analysis

There are obvious differences in resource endowment, institutional conditions, and development levels among various regions in China. Therefore, the impression of HSR construction on green technology innovation may not be consistent in all regions and cities with the same degree and sensitivity. Sub-sample regression is needed to test whether it is consistent with the main conclusion. Given the defect of generating the sub-samples in the eastern, middle, and western regions, this paper will classify the samples according to locational conditions and marketization degree.

Firstly, this paper uses circulation node cities, which are defined in National Distribution Planning for Circulation Node Cities (2015–2020) [103] as the division basis of the difference in locational conditions. The samples were divided into circulation node cities (including national and regional) and other cities. Cities selected as circulation nodes are in a key and important position in the national distribution network. They have advantages in basic conditions, such as business flow, logistics flow, capital flow, and information flow. The findings of Columns (1) to (2) in Table 4 demonstrate that circulation node cities benefit from the opening of HSRs. This may be due to the accumulation of green innovation elements in circulation node cities to some extent. The opening of HSRs further promotes the integration and utilization of resources, thus significantly advancing the output of green technology innovation. In contrast, for other cities, although the flow capacity has improved after integration into the HSR network, the level of green technology innovation still needs time to improve.

Secondly, the heterogeneity of HSR impact is discussed from the perspective of marketization degree. Referring to Fan et al. [104] and Li et al. [105], we calculate the marketization index of China’s prefecture-level cities in 2018. The samples are classified in line with the standard of 2018. The city whose marketization index exceeds the median (12.8726) is classified as having high marketization, whereas the city whose marketization index is lower than the median is low marketization. As disclosed in Columns (3) and (4) of Table 4, the coefficient of HSRs is suggestively positive (at 1% confidence level) in the city group with a low degree of marketization but not meaningful in the city group with a high mark of marketization. This seems to suggest that cities with a low degree of marketization obtain a stronger marginal effect of HSRs. Conversely, cities with a high degree of marketization have their own advantages in green technology innovation [106]; so the opening of HSRs has no momentous impression.

### 5.4. Mechanism Analysis

Owing to mechanism analysis, this paper argues that the establishment of HSRs mainly affects the green technology innovation of cities through the creative class flow and the government’s environmental concern. Therefore, this paper adds mechanism variables and their interaction terms with HSRs to the basic regression for testing whether relevant mechanisms are established. Note that mechanism variables are centralized before multiplying. Table 5 displays the findings of the mechanism analysis based on the panel negative binomial model (NB).

Column (1) in Table 5 adds the variable of creative class flow (CCF) into the basic regression. The outcomes show that the increase in creative class flow will significantly improve the output of green technology innovation. This complies with the findings of Huang and Wang [13]. Column (2) adds the interaction term (HSR_CCF) between HSRs and creative class flow. Its coefficient is momentously negative at the confidence level of 5% and is opposite to the HSRs’ coefficient. This indicates that the marginal effect of HSRs on green technology innovation will be changed with the intensity of CCF. When CCF is at a low value, the marginal influence of HSRs is positive. However, when CCF is at a high level, the marginal effect of HSRs is negative. Namely, the opening of HSRs can suggestively improve the green technology innovation ability of cities with a poor flow of the creative class. As analyzed above, the opening of HSRs can accelerate the dissemination and generation of knowledge. We believe that HSRs can help low-mobility cities to access resources related to green technology innovation from high-mobility cities. Meanwhile, low-mobility cities will also face greater market competition, which will lead to green technology innovation.

Column (3) in Table 5 illustrates the outcomes of adding the variable of environmental concern (EC). The coefficient of EC is positive, but the substance is poor. We find that the introduction of HSRs can remarkably advance the output of green technology innovation, but the government’s increased environmental concern does not stimulate green technology innovation. There are probably two reasons. On the one hand, it may be because the government’s heed to the environment stops at the level of formulating implementation documents but does not take action. On the other hand, it may be because the government’s implementation of multiple combinations of environmental regulations forms opposite forces on green technology innovation, which cancel each other out. Hence, EC fails to play a significant promoting role [107]. However, we find that the coefficient of the interaction term (HSR_EC) in Column (4) is considerably positive. It means that the marginal effect of HSRs on green technology innovation will be moderated by the government’s environmental concerns. When the value of EC is higher, the endorsement effect of HSRs on green technology innovation is greater. Although the government’s strong concern for the environment has not significantly stimulated green technology innovation in cities, the opening of HSRs could make it visible. As previously discussed, HSRs can inspire governments to take real action to improve the environment. As a result, the introduction of HSRs will finally stimulate green technology innovation through the government’s environmental concerns.

### 5.5. Spatial Spillover Effect

Before utilizing the spatial econometric model to assess the spillover influence, it is essential to examine the spatial correlation of the samples. Therefore, the global and local Moran indices of the green patent are calculated. Table 6 displays the outcomes of the global Moran index for all years of green patents. The global Moreland index was positive at the 1% confidence level for almost all years, and the value increased with the year. It means that green technology innovation has a positive spatial correlation, and the correlation is becoming more and more obvious. In addition, the local Moreland index scatter plots can reflect the spatial agglomeration features of green innovation. Due to the word limit, we only highlight the scatter plots for three years. Figure 2, Figure 3 and Figure 4 show the local Moreland index scatter plots of green patents in 2007, 2012, and 2018, respectively, reflecting the dynamic revolution of spatial agglomeration characteristics during the study period. The horizontal axis denotes the number of standardized green patent applications, and the vertical axis signifies the spatial lag value. it has been observed that a large number of cities are located in the first and third quadrants, revealing the agglomeration of similar values. It proves that green technology innovations are positively spatially correlated. Peng et al. [108] and Zhou et al. [109] hold similar views.

Column (1) of Table 7 displays the regression outcomes of SDM founded on Formula 4. Meanwhile, for robustness, the estimation results of the Spatial Lag Model (SLM) and the Spatial Error Model (SEM) are also reported in Columns (2) and (3). The coefficients of HSRs and $W_{1}$HSRs in Column (1) are considerably positive at the confidence level of 10% and 5%, respectively. Hence, we can preliminarily confirm that the opening of HSRs promotes green technology innovation in the local and adjoining regions. That is, HSRs maintain a momentous spatial spillover impression. Furthermore, the spatial lag term of green technology innovation (rho) is appreciably positive (at the 1% confidence level) in both Columns (1) and (2). It means that the intensity of green technology innovation in adjoining areas influence each other. The spatial lag term of the error term (lambda) in Column (3) is also positive at the 1% level, revealing that the spatial correlation also comes from the disturbance term. In addition, the likelihood-ratio test (LR) was performed to confirm if the SDM model is required to be degraded to SLM or SEM. The outcomes disclose that the null hypothesis is rejected under the condition of 1%; that is, the SDM model is still the best choice. The outcomes of the Hausman test also confirm the correctness of using the fixed effect model.

However, the spatial spillover effects of variables cannot be directly evaluated by the coefficients of explanatory variables and their spatial lag terms in Column (1) of Table 7. Therefore, the direct, indirect, and total effects of each explanatory variable on local green technology innovation are further reported in Table 8. As displayed in Table 8, the direct effect of HSRs is 0.088 at the 5% confidence level. This denotes that the introduction of HSRs could directly lead to an 8.8% increase in local green technology innovation. This is in compliance with the deduction of baseline regression, which again confirms Hypothesis 1 (H1). The indirect effect of HSRs was shown to be 0.425 with a 1% confidence level. It means that the opening of HSRs in nearby cities could also lead to a 42.5% rise in local green technology innovation. This complies with the conclusions of Jiao et al. [82] and Yu et al. [49]. It proves that HSRs in geographically close cities have a noticeable positive spatial spillover influence on local green technology innovation instead of the Siphon effect. The total effect is the sum of the direct and indirect effects.

## 6. Conclusions

As an important representative of green transportation in China, HSRs plays an indispensable role in China’s efforts toward green development and dual carbon goals. Using panel data from 286 Chinese cities between 2007 and 2018, this paper explores the impact of HSRs on green technology innovation and its mechanism. For ensuring the validity of the conclusion, this paper not only uses three methods to deal with endogeneity problems but also carries out a range of robustness tests. The heterogeneity is analyzed from perspectives of locational conditions and marketization degree. Finally, the Spatial Dubin Model (SDM) is applied to further measure the influence of HSRs on green technology innovation in surrounding cities.

This paper principally draws the following conclusions. Firstly, the development of HSRs has a direct and substantial advancement effect on green technology innovation. Secondly, in terms of location conditions, the opening of HSRs plays a role in endorsing green technology innovation for circulation node cities. However, for other ordinary cities, the effect is not obvious. From the standpoint of marketization degree, the marginal influence of HSRs on green technology innovation is stronger in cities with lower marketization degrees. Next, HSRs can indirectly stimulate the expansion of green technology innovation by influencing the creative class flow and environmental concerns of the local government. Finally, the stimulation of HSRs on green technology innovation also has a remarkable positive spatial spillover impact.

This paper finds that HSRs contribute to green technology innovation and can increase the number of green patent applications in cities. It confirms that building the HSR network, as an important part of the government’s expansion of a modern integrated transportation system, is a key means to accelerate the development of green technology innovation. Therefore, China should continue to improve the layout and construction of the HSR network and build a convenient, smooth, economic, and efficient urban corridor.

Since the mobility of the creative class is an essential way for HSRs to influence green technology innovation, local governments should further remove obstacles to the flow of talent while developing HSRs. Local governments can improve their ability to absorb talent not only by offering better talent introduction plans but also by improving the livability of local cities. Meanwhile, HSRs have significant spatial spillover benefits. So, the circulation node cities connected by HSRs should draw full attention to the influence of green technology innovation. They should effectively use the resources, such as talent, capital, and knowledge gathered, and speed up the innovation and transformation of green technology. Additionally, they should play an exemplary and leading role and share successful green technology innovation models promptly. Thus, the whole society is driven to the achievement of green development.

As stated in the findings, the introduction of HSRs will bring a variety of green innovation resources to local governments, and, the environmental concerns of local governments will affect the output of green technology innovation. Therefore, for cities that have the HSRs, local governments should attach importance to the harmonious advance of the economy and environment. They can introduce incentive policies related to green innovation. For cities that have not yet opened HSRs, local governments should strive to establish local stations for HSR lines. Meanwhile, they should use environmental protection policies, fiscal and tax policies, educational and medical resources, and other programs to attract the inflow of production factors. These policies would improve the competitiveness of cities.

However, there are still some deficiencies in this paper. As China’s HSRs are still developing rapidly, the data used in the empirical part are relatively outdated and fail to keep up with the actual level. It may lead to some deviations between research results and reality. Moreover, this paper uses the number of green patent applications as the proxy variable of green technology innovation, which lacks the consideration of the input–output efficiency of green technology innovation. Hence, this study still needs to be further supplemented and improved.

## Figures and Tables

**Figure 1 ijerph-20-00347-f001:**
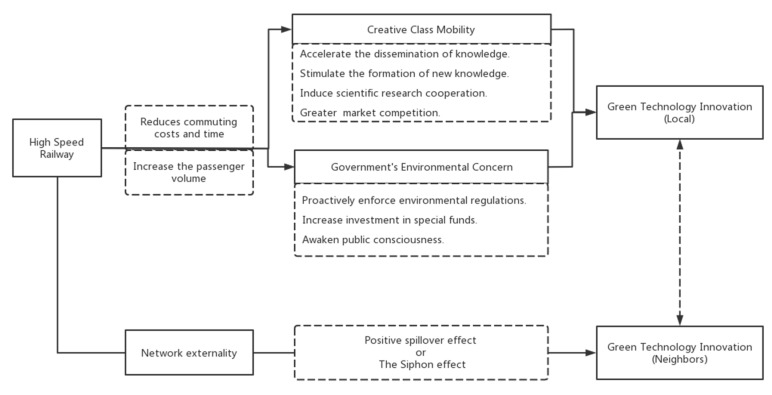
The conceptual diagram.

**Figure 2 ijerph-20-00347-f002:**
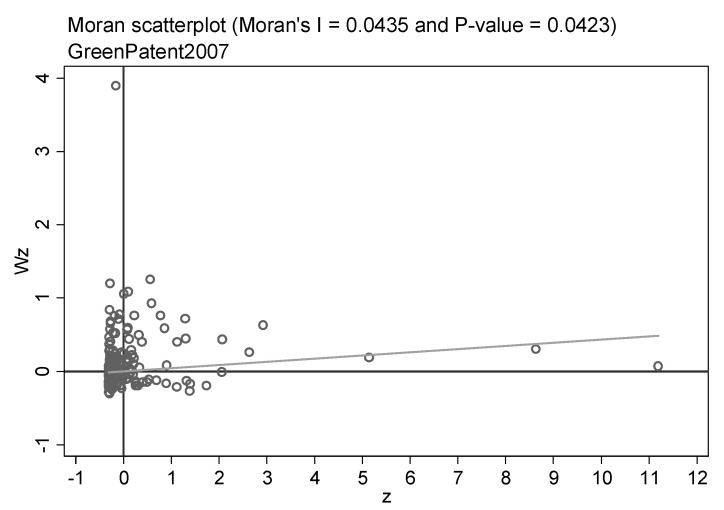
Scatter plots of the local Moran’s index of green patents in China’s cities in 2007.

**Figure 3 ijerph-20-00347-f003:**
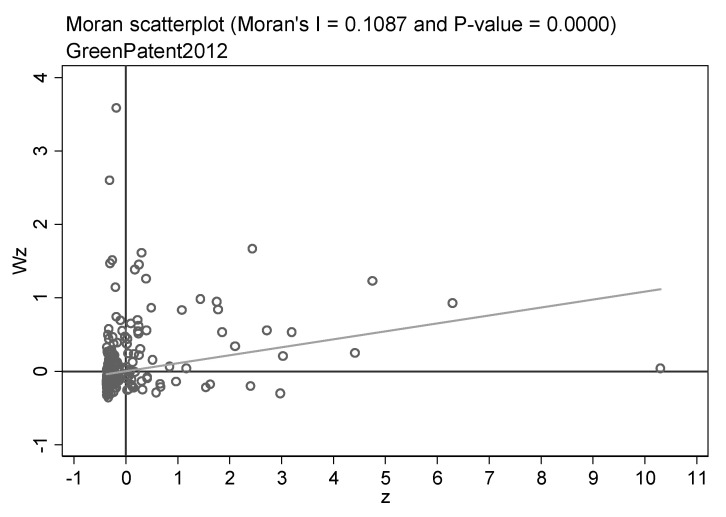
Scatter plots of the local Moran’s index of green patents in China’s cities in 2012.

**Figure 4 ijerph-20-00347-f004:**
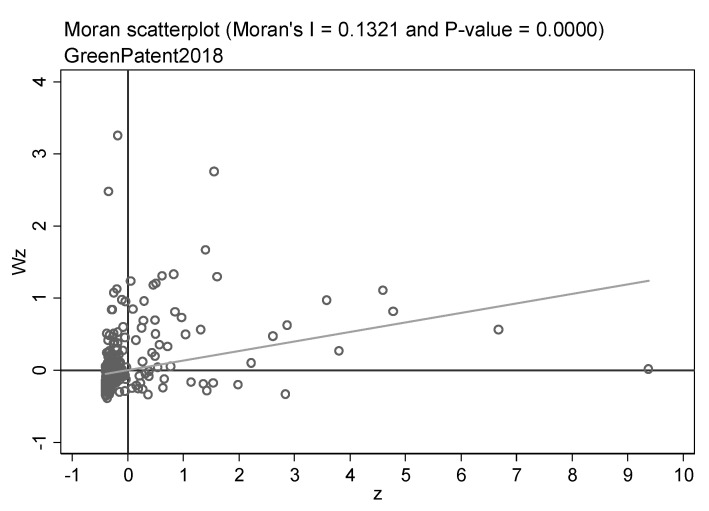
Scatter plots of the local Moran’s index of green patents in China’s cities in 2018.

**Table 1 ijerph-20-00347-t001:** Descriptive statistics of main variables.

	Variable	Obs	Mean	Std. Dev.	Min	Max
Dependent Variable	GTI	3432	379.807	1232.033	0	21,197
Independent Variable	HSRs	3432	0.462	0.499	0	1
Control Variables	ED	3432	10.433	0.732	0	13.056
	IS	3432	0.822	0.250	0.190	2.452
	GS	3432	0.002	0.002	0	0.061
	FD	3432	1.120	0.296	0	2.676
	FI	3432	1.351	0.784	0	3.832
	otherVol	3432	9.074	2.038	0	18.094
Mechanism Variables	CCF	3432	6.523	1.298	0	12.132
	EC	3432	2.967	0.558	0	4.543

**Table 2 ijerph-20-00347-t002:** Baseline regression and robustness test.

	(1)	(2)	(3)	(4)	(5)	(6)
Variables	NB	NB	NB	Poisson	FE	Exclude
HSRs	0.076 **	0.072 **		0.108 **	0.066 *	0.060 **
	(2.14)	(2.18)		(2.05)	(1.66)	(2.25)
L1HSRs			0.072 **			
			(2.21)			
ED		0.413 ***	0.414 **	0.085	0.130 ***	0.269 *
		(2.59)	(2.54)	(0.83)	(2.91)	(1.85)
IS		0.194 *	0.199 *	0.097	0.369 ***	0.247 **
		(1.74)	(1.76)	(0.37)	(2.67)	(2.02)
GS		0.050 ***	0.047 ***	0.061 ***	0.053 ***	0.063 ***
		(3.90)	(3.83)	(4.09)	(5.03)	(5.85)
FD		0.220	0.238	−0.140	−0.043	0.004
		(1.56)	(1.61)	(−0.73)	(−0.32)	(0.02)
FI		0.006	0.016	−0.040	−0.037	0.012
		(0.23)	(0.63)	(−0.75)	(−1.19)	(0.44)
otherVol		0.013	0.013	0.030 ***	0.009	0.017
		(1.46)	(1.58)	(2.72)	(1.11)	(1.57)
Constant	0.634 ***	−3.566 **	−3.556 **		1.568 ***	−2.017
	(9.18)	(−2.23)	(−2.14)		(3.13)	(−1.34)
Observations	3432	3432	3146	3432	3432	3060
Number of cities	286	286	286	286	286	255
City FE	Yes	Yes	Yes	Yes	Yes	Yes
Time FE	Yes	Yes	Yes	Yes	Yes	Yes
Log likelihood	−14,644	−14,524	−13,437	−40,529		−12,342
Wald Chi2 (*p*-value)	7622 (0.000)	9825 (0.000)	7909 (0.000)	6239 (0.000)		7992 (0.000)
R-squared					0.824	

Note: ***, **, and * indicate significance at the 1%, 5%, and 10% levels, respectively.

**Table 3 ijerph-20-00347-t003:** Endogenous problem.

	(1)	(2)
Variables	IV-GMM	SysGMM
HSRs	0.665 ***	0.025 *
	(3.70)	(0.66)
L.lnGP		0.822 ***
		(14.60)
L2.lnGP		0.136 ***
		(2.75)
Observations	3432	2860
R-squared	0.794	
Number of cities	286	286
Control variables	Yes	Yes
City FE	Yes	Yes
Time FE	Yes	Yes
Under identification test	0.000	
Weak identification test	17.57	
Hansen J statistic	0.455	
AR(1)		0.000
AR(2)		0.389
Hansen test		0.109
Wald Chi2 (*p*-value)		717,214 (0.000)

Note: *** and * indicate significance at the 1% and 10% levels, respectively.

**Table 4 ijerph-20-00347-t004:** Heterogeneity analysis.

	(1)	(2)	(3)	(4)
Variables	Node City	Not N City	High Marketization	Low Marketization
HSRs	0.101 *	0.041	0.004	0.153 ***
	(1.91)	(0.96)	(0.12)	(3.03)
Constant	0.156	−4.356 ***	−4.058 *	−3.186 **
	(0.10)	(−3.58)	(−1.87)	(−2.14)
Observations	1128	2304	1716	1716
Number of cities	94	192	143	143
Control variables	Yes	Yes	Yes	Yes
City FE	Yes	Yes	Yes	Yes
Time FE	Yes	Yes	Yes	Yes
Log likelihood	−5736	−8753	−7301	−7204
Wald Chi2 (*p*-value)	2754 (0.000)	4943 (0.000)	5671 (0.000)	4007 (0.000)

Note: ***, **, and * indicate significance at the 1%, 5%, and 10% levels, respectively.

**Table 5 ijerph-20-00347-t005:** Mechanism analysis.

	(1)	(2)	(3)	(4)
Variables	NB	NB	NB	NB
HSRs	0.047	0.340 **	0.072 **	0.069 **
	(1.42)	(2.37)	(2.19)	(2.05)
CCF	0.205 ***	0.228 ***		
	(4.95)	(4.98)		
HSR_CCF		−0.047 **		
		(−2.09)		
EC			0.011	−0.014
			(0.64)	(−0.60)
HSR_EC				0.047 *
				(1.65)
Constant	−3.874 ***	−4.081 ***	−3.571 **	−3.486 **
	(−4.59)	(−4.59)	(−2.38)	(−2.17)
Observations	3430	3430	3432	3432
Number of cities	286	286	286	286
Control variables	Yes	Yes	Yes	Yes
City FE	Yes	Yes	Yes	Yes
Time FE	Yes	Yes	Yes	Yes
Log likelihood	−14,426	−14,421	−14,524	−14,522
Wald Chi2 (*p*-value)	10,603 (0.000)	10,823 (0.000)	9974 (0.000)	10,125 (0.000)

Note: ***, **, and * indicate significance at the 1%, 5%, and 10% levels, respectively.

**Table 6 ijerph-20-00347-t006:** Global Moran’s index of green patents.

Year	I	E(I)	Sd(I)	Z	*p*-Value
2007	0.0435	−0.0035	0.0232	2.0309	0.0423
2008	0.0458	−0.0035	0.0223	2.2151	0.0268
2009	0.0543	−0.0035	0.0234	2.4675	0.0136
2010	0.0837	−0.0035	0.0239	3.6481	0.0003
2011	0.0950	−0.0035	0.0241	4.0900	0.0000
2012	0.1087	−0.0035	0.0247	4.5498	0.0000
2013	0.0791	−0.0035	0.0229	3.6100	0.0003
2014	0.0817	−0.0035	0.0224	3.7971	0.0001
2015	0.1028	−0.0035	0.0238	4.4768	0.0000
2016	0.1118	−0.0035	0.0245	4.6992	0.0000
2017	0.1196	−0.0035	0.0251	4.9114	0.0000
2018	0.1321	−0.0035	0.0252	5.3811	0.0000

**Table 7 ijerph-20-00347-t007:** Spatial spillover effects.

	(1)	(2)	(3)
Variables	SDM	SLM	SEM
HSRs	0.074 *	0.199 ***	0.113 ***
	(1.92)	(5.64)	(2.88)
ED	0.139 ***	0.319 ***	0.157 ***
	(3.97)	(10.17)	(4.30)
IS	0.161	−0.259 ***	0.069
	(1.48)	(−2.79)	(0.62)
GS	0.025 **	0.039 ***	0.018
	(2.08)	(3.35)	(1.52)
FD	0.023	0.424 ***	0.150
	(0.19)	(4.04)	(1.20)
FI	−0.067 **	−0.084 ***	−0.081 ***
	(−2.22)	(−3.18)	(−2.67)
otherVol	0.017 **	−0.011	0.016 **
	(2.20)	(−1.51)	(1.97)
$W_{1}$HSRs	0.235 **		
	(2.57)		
$W_{1}$ED	0.466 ***		
	(6.49)		
$W_{1}$IS	−0.695 ***		
	(−3.60)		
$W_{1}$GS	0.096 ***		
	(3.26)		
$W_{1}$FD	0.827 ***		
	(3.34)		
$W_{1}$FI	−0.055		
	(−0.89)		
$W_{1}$otherVol	−0.114 ***		
	(−6.49)		
rho	0.403 ***	0.667 ***	
	(13.42)	(34.14)	
lambda			0.855 ***
			(68.06)
sigma2_e	0.337 ***	0.344 ***	0.346 ***
	(41.11)	(40.85)	(40.64)
Observations	3432	3432	3432
R-squared	0.444	0.552	0.542
Log likelihood	−3033.136	−3138.268	−3236.718
Number of cities	286	286	286

Note: ***, **, and * indicate significance at the 1%, 5%, and 10% levels, respectively.

**Table 8 ijerph-20-00347-t008:** Direct, indirect, and total effects based on the results of Model (1) in Table 7.

Variables	HSR	ED	IS	GS	FD	FI	Othervol
Direct effect	0.088 **	0.164 ***	0.141	0.030 ***	0.068	−0.070 **	0.012
	(2.29)	(4.91)	(1.39)	(2.60)	(0.59)	(−2.34)	(1.46)
Indirect effect	0.425 ***	0.856 ***	−1.026 ***	0.170 ***	1.373 ***	−0.139	−0.172 ***
	(3.08)	(9.33)	(−3.54)	(3.71)	(3.67)	(−1.55)	(−6.18)
Total effect	0.513 ***	1.020 ***	−0.884 ***	0.200 ***	1.441 ***	−0.209 **	−0.160 ***
	(3.74)	(10.85)	(−3.09)	(4.25)	(3.89)	(−2.24)	(−5.42)

Notes: *** and ** represent significance levels of 1% and 5%, respectively.

## Data Availability

The data that support the findings of this study are available on request from the corresponding author. The data are not publicly available due to privacy or ethical restrictions.

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
