# Peer review of "A Key to Stimulate Green Technology Innovation in China: The Expansion of High-Speed Railways"

_ijerph, 2022, doi:10.3390/ijerph20010347_

Round 1

Reviewer 1 Report

Thank you for the opportunity to review this work. The paper addresses a very interesting topic, that is, if the high-speed railway, as a typical representative of green transportation, can effectively promote green technology innovation in cities. To do this, the paper uses the panel data of Chinese prefecture-level cities from 2007 to 2018 for empirical analysis.

The paper’s argument is built on an appropriate base of theory, and the research is well designed.  In this case the statement of objectives such as filling the existing gap in research that does not take into account the actions of local governments or lacks rigorous empirical support, between others, are correct for the resolution of the hypothesis. 

The introduction and the literature review covers a lot of background material, it demonstrates an adequate understanding of the relevant literature in the field and cite an appropriate range of literature sources. The first paragraphs point out a very important idea, that few scholars have explored its incentive force on green technology innovation from the perspective of transportation expansion. It is well argued but it is recommended to review the writing so that it is clearer, and to translate the paragraphs indicating that they are translations made by the authors.

Regarding methodology, in the section 4, it would be advisable to clarify at the beginning of the section the type of methodology used before analyzing the variables.

This manuscript signalizes that HSR contributes to green technology innovation and can increase the amount of green patent applications in cities. It confirms that building the HSR network, as an important part of the government's expansion of a modern integrated

transportation system. The results show an interesting perspective despite the deficiencies pointed out in the conclusions. Perhaps in the future the authors can get funding to help them alleviate them.

This is an interesting and valuable discussion, I think that this manuscript is well-researched and an important contribution to knowledge. I recommend a review of the paper by the authors in relation to the comment made by the reviewer, after which the manuscript will be ready for publication.

Reviewer 2 Report

This interesting study is well constructed in general. The problem, assumptions, and results are well characterized. My basic remark concerns the associations with similar research described in the literature.

The reference list should be enhanced considering the present positions from IJERPH journal (and MDPI group). Connections of presented research with the IJERPH topics should be indicated (in the introduction, literature review, and in conclusions).

Please, use your own keywords: high-speed railways; green technology innovation; creative class flow; environmental concerns; spatial spillover effect to find representative literature sources.

For example, the keywords “high-speed railways” resulted in 13 papers only in IJERPH journal, and the keywords “green technology innovation” – 101 in the same publisher.

The paper considers the data from 2007 to 2018 (rows 43, 313, table 6, etc.). Why not from later years? Is a Covid-19 effect considered? These aspects should be clarified.

Please, add the basic information about the Chinese HSR system.

I don’t like the structure of sections 4.1.3 and 4.1.4. The better form will be with an introduction like “We developed (or use) six control variables called …” and next the description of them.

Table 1: is a such big number of digits sensible? For example, 9.074412 should be replaced by 9.074. Use zero before dotting in this table.

Please, divide section 5 (Analysis of Findings) into two. The first one called “results” should present the effect of the research with no commentaries. The second section called “discussion” (before “conclusions”) will concern the general commentary on results and the comparison with the literature.

I think, that the preciously referring to the theses formulated in section 3 (with connections to their numbers) in the conclusions will be nice.

Some less important and technical remarks:

-       I see in the head of pages the year 2021,

-       reference system must be corrected (according to rules of IJERPH journal),

-       contribution of all Authors must be shown (pages 17 and 18),

-       construction of tables, their position on pages, and division into pages should be improved.

Reviewer 3 Report

Reviewer Comments

The authors have written on ‘The association between high-speed rail and green technology innovation in China”. I believe the readers would find the article more interesting if most of the sentences in the introduction were appropriately linked to each other. I suggest the service of a language editor to ensure that the manuscript is free from any form of grammatical error. In addition to previously stated observations, the following suggestions should help improve the quality of the manuscript:

Abstract

1. In The first line of the abstract, can the authors find another word for “allowances”

2.     Why are the authors focusing only on 2007 to 2018?

3.     In the last lines of the abstract, the authors should further indicate what is or are the significant contribution of their work to the field of green transportation and high-speed railway.

4.     The authors should re-write the methodology in the abstract section. The methodology section in the abstract is not convincing enough to convince a first reader about what they used in this research.

5.     I noticed that the authors lay emphasis on “spatial spillover effect” a lot throughout the research. Can the authors find a way to add it to the title of the manuscript?

Introduction

1.     First of all, the introduction is well-written, and it justifies the research, however, can the authors add a little bit more to buttress the introduction section?

Literature Review

1.     The literature review is okay, but it still needs improvement. The authors should focus on research done within the last seven years at most in this field and cite them. It is good to cite old articles, but it is important to limit the citation of old articles; this is 2021. Just searching their topic or research on google scholar, I can see a lot of research on it in the last five years. The authors need to include the literature review in their research.

Methodology

1.     A flowchart needs to be introduced at the beginning of the methods and data section in order for readers to understand the conceptual and methodological framework of this research just by looking at the flowchart.

2.     Can the authors introduce a table showing the dependent, independent, control and mechanism variable?

3.     The authors should introduce a table showing the sample of the data.

4.     How was the data extracted? Where is the data used for this research?

5.     The author needs to go into more detail and explain step-by-step on how they use “baseline regression model” to estimate the connection between HSR and the green technology innovation.

Round 2

Reviewer 2 Report

All of my remarks were considered in a general.

I identify some smaller problems now:

-       subsection 4.3 is in the inappropriate place (page 5),

-       it is still the publication year for this paper 2021, this year is done, and publication will be probably in 2023,

-       the answer “Given the remarkable impact of the pandemic of Covid-19, the research period is set as 2007 to 2018” should be not only sent to the reviewer but also composed into the text,

-       do the Author not know that this is not a double-blind review?

Reviewer 3 Report

Dear Authors,

I am satisfied with the answers to the reviewer's comments.

Author Response

Many thanks for your suggestions.